# A Coarse-to-Fine Fusion Network for Small Liver Tumor Detection and Segmentation: A Real-World Study

**DOI:** 10.3390/diagnostics13152504

**Published:** 2023-07-27

**Authors:** Shu Wu, Hang Yu, Cuiping Li, Rencheng Zheng, Xueqin Xia, Chengyan Wang, He Wang

**Affiliations:** 1Zhiyu Software Information Co., Ltd., Shanghai 200030, China; 2Institute of Science and Technology for Brain-Inspired Intelligence, Fudan University, Shanghai 200433, China; 3Human Phenome Institute, Fudan University, Shanghai 200433, China; 4Department of Neurology, Zhongshan Hospital, Fudan University, Shanghai 200032, China; 5Key Laboratory of Computational Neuroscience and Brain-Inspired Intelligence (Fudan University), Ministry of Education, Shanghai 200433, China

**Keywords:** dynamic contrast-enhanced imaging, segmentation, lesion detection, small liver tumor, convolutional neural network, deep learning

## Abstract

Liver tumor semantic segmentation is a crucial task in medical image analysis that requires multiple MRI modalities. This paper proposes a novel coarse-to-fine fusion segmentation approach to detect and segment small liver tumors of various sizes. To enhance the segmentation accuracy of small liver tumors, the method incorporates a detection module and a CSR (convolution-SE-residual) module, which includes a convolution block, an SE (squeeze and excitation) module, and a residual module for fine segmentation. The proposed method demonstrates superior performance compared to conventional single-stage end-to-end networks. A private liver MRI dataset comprising 218 patients with a total of 3605 tumors, including 3273 tumors smaller than 3.0 cm, were collected for the proposed method. There are five types of liver tumors identified in this dataset: hepatocellular carcinoma (HCC); metastases of the liver; cholangiocarcinoma (ICC); hepatic cyst; and liver hemangioma. The results indicate that the proposed method outperforms the single segmentation networks 3D UNet and nnU-Net as well as the fusion networks of 3D UNet and nnU-Net with nnDetection. The proposed architecture was evaluated on a test set of 44 images, with an average Dice similarity coefficient (DSC) and recall of 86.9% and 86.7%, respectively, which is a 1% improvement compared to the comparison method. More importantly, compared to existing methods, our proposed approach demonstrates state-of-the-art performance in segmenting small objects with sizes smaller than 10 mm, achieving a Dice score of 85.3% and a malignancy detection rate of 87.5%.

## 1. Introduction

According to the World Health Organization (WHO), liver cancer ranks as one of the most prevalent forms of cancer globally, resulting in a significant number of fatalities annually [1]. Hepatocellular carcinoma (HCC), the most common primary liver cancer type, stands as the fifth most widespread malignancy and the third leading cause of cancer-related death worldwide. In general, accurate segmentation of the liver and liver tumor is an important prerequisite before surgery to help physicians make accurate assessments and treatment plans. Traditionally, liver and liver-tumor segmentation rely on manual annotation by radiologists [2], which is time-consuming and susceptible to personal subjective experience. Therefore, there is an urgent need for automated liver and tumor segmentation methods for healthcare professionals in clinical practice. Recently, deep learning has achieved remarkable results in liver tumor segmentation, and most of the top-ranked methods in the 2017 Liver Tumor Segmentation (LiTS) Challenge were based on deep learning [3]; however, the accurate automatic segmentation of small liver tumors is still very challenging.

In complex and diverse real-world scenarios, detection and segmentation of large targets often work well but are not satisfactory for small targets such as early tumors and vascular plaque [4]. The main difficulties for the accurate detection and segmentation of small objects in medical images are as follows: small object region to be detected; few extractable features for small objects; and susceptibility to noise interference [5]. At present, there is still relatively little research dedicated to these problems, so it is an important research direction to explore how to improve the mainstream detection and segmentation algorithms to make them effective for small object detection and the segmentation of medical images [6]. Research on small object detection will help promote the development of the target detection field, broaden the application scenarios of target detection in the real world, improve the level of scientific and technological innovation, and accelerate the pace of the world’s overall step into an era of intelligence [7]. Though further fine segmentation may bring more value, and quantifying small objects may once again promote the progress of AI in medicine [8], little research has combined small object detection and segmentation for medical images, and most of the research almost only does detection, for example, lung-nodule detection in medicine [9].

In this paper, we propose a multi-network fusion segmentation framework to comprehensively detect and segment benign and malignant liver tumors and, in particular, to improve the performance of the model on small tumor segmentation. The end-to-end network is more difficult to detect and segment tumors comprehensively as liver tumors vary in size and types of features. Whereas multi-network fusion and coarse-to-fine segmentation can compensate for the shortcomings of end-to-end networks in this regard, the proposed method can learn features and fuse tumors in high and low dimensions of the image; moreover, the fusion network combines detection and segmentation to complement each other to improve the detection rate and segmentation performance simultaneously. The proposed model was evaluated on an MRI liver tumor dataset containing different sizes and different types of liver tumors, and we innovatively provided evaluation methods for this fusion method in terms of detection and segmentation metrics. Our proposed multi-stage coarse-to-fine fusion segmentation method refines the coarse segmentation results at different stages of the network architecture and innovatively includes a detection module as well as a CSR module—consisting of a convolutional block, SE module, and residual module—at the tumor fine segmentation stage. For specific information on the CSR module and SE module, see the CSR-UNet paragraph. The proposed method shows superior performance compared to the segmentation capability of single-stage end-to-end network architecture.

### Related Work

To improve the tumor segmentation efficacy on MRI, much of the related research is directed at the improvement of neural network structures or related parameters. Jin et al. [10] proposed a 3D hybrid residual attention-aware segmentation method, i.e., RA-UNet, to precisely extract the liver region and segment tumors from the liver. Two-dimensional convolutions cannot fully leverage the spatial information along the third dimension, while 3D convolutions suffer from high computational costs and high GPU memory consumption. Although deep convolutional neural networks (DCNNs) have contributed to many breakthroughs in image segmentation, the task still remains challenging since 2D DCNNs are incapable of exploring the inter-slice information and 3D DCNNs are too complex to be trained with the available small dataset. Tang et al. [11] proposed a two-stage framework for 2D liver and tumor segmentation. Umer et al. [12] applied a simpler and faster one-stage detector RetinaNet for the localization of liver tumors on LiTS17, and the proposed method precisely detects one or more tumors. Ayalew et al. [13] propose a liver and tumor segmentation method using a UNet architecture as a baseline. Due to the heterogeneity and low contrast of biomedical images, current state-of-the-art tumor-segmentation approaches are facing the challenge of the insensitive detection of small tumor regions. To tackle this problem, Wong et al. [14] proposed a network architecture and the corresponding loss function, which improved the segmentation of very small structures. Kofler et al. [15] proposed a novel family of loss functions, nicknamed blob loss, primarily aimed at maximizing instance-level detection metrics, such as F1 score and sensitivity. Isensee et al. proposed the nnU-Net [16], which as a benchmark medical image segmentation method provides 2D and 3D models based on U-Net, and also provides a variety of image preprocessing and enhancement methods.

In addition, some multi-stage deep network models have been proposed and proven to be effective. Li et al. [17] proposed a new three-stage curriculum learning approach for training deep networks to tackle this small object segmentation problem. It is a challenging task since the tumors are small against the background. The experimental results show that compared with the traditional U-Net, the Dice index of liver and tumor segmentations with the improved model increased by 5.14% and 2.63%, respectively, and the recall rate increased by 1.8% and 9.05% [18]. Luan et al. [19] proposed a neural network (S-Net) that can incorporate attention mechanisms for end-to-end segmentation of liver tumors from CT images. Li et al. [20] proposed a Liver and Tumor Segmentation Network (LiTS-Net) framework. Yang et al. [21] aimed to improve liver tumor detection performance by proposing a dual-path feature-extracting strategy and employing Swin Transformer. Fan et al. [22] provided a multiscale nested U-net (MSN-UNet) for liver segmentation. The MSN-Unet contains multiscale context fusion (MSCF) blocks that acquire multiscale semantic data and obtain multilevel feature maps, and the Res block exploits residual connectivity. Tang et al. [23] proposed an enhanced region CNN (R-CNN) and DeepLab for liver segmentation. The deep learning method was applied to the detection and segmentation steps of the liver to reduce the influence of human factors on the segmentation. The principle of residual learning is also utilized to fuse and extract multi-level information from the network using a jump structure.

## 2. Materials and Methods

### 2.1. Data

#### 2.1.1. Patient Inclusion

We developed and evaluated our model based on a private dataset from Shanghai Public Health Clinical Center (Affiliated with Fudan University) and Shanghai ZhiYu Software Technology Co., Ltd., Shanghai, China. This dataset has a total of 218 abdominal MRI scans of liver cancer patients, and we used the sequence of DCE imaging delayed phase. This data set included 100 women and 118 men with an average age of 45 years. The liver tumor categories were Hepatocellular carcinoma (HCC), metastases of liver, Cholangiocarcinoma (ICC), hepatic cyst, and liver hemangioma.

#### 2.1.2. Dataset

MRI was performed on a 3.0 T clinical scanner (Ingenia, Philips Medical System). Gadopentetate dimeglumine (Magnevist; Bayer Healthcare, Leverkusen, Germany, 0.1 mmol kg^−1^) was injected at a rate of 2 mL s^−1^ followed by a saline flush with a maximum volume of 20 mL. The images of hepatic arterial, portal, and delay phases were obtained at 25–30 s, 60–90 s, and 180 s after contrast medium injection, respectively.

The recorded pixel sizes of our datasets were 640 × 640 × 60 corresponding to a spatial resolution of 0.5938 × 0.5938 × 5.0 mm^3^. The main target of our study was detection and segmentation of small liver tumors, and the main object of this study was the differentiation of space-occupying tumors with long diameters from 5 mm to 30 mm and tumor category from benign to malignant.

The dataset was labeled by Shanghai ZhiYu Software Technology Co., Ltd., according to the category of the tumor, and the distribution of liver tumors in terms of size and type as shown in Figure 1. Specifically, a total of 3605 liver tumors were counted, of which the type occupying the major proportion was hepatocellular carcinoma. The benign types were hepatic cysts and liver hemangioma, and the malignant types were HCC, metastases of liver, and ICC, with more than half of the malignant types. For tumor size, more than 50% of the tumors were between 10 and 30 mm, while a slightly lower percentage were between 5 and 10 mm. Since the slice thickness of MRI was more than 5 mm, it was not possible to quantify the occupancies smaller than 5 mm, so they were classified as 5–10 mm in this paper. Our main subject, small liver tumors smaller than 30 mm, occupied the majority of this dataset with 90.7%. Some delay phases with their ground truth are shown in Figure 2.

### 2.2. Methods

#### 2.2.1. Architecture

The model architecture mainly includes three steps: preprocessing and liver segmentation; tumor detection by nnDetection model; coarse-to-fine segmentation through CSR-UNet and merging, shown in Figure 3.

Firstly, in the preprocessing process, each slice of 3D MRI data was resampled into a size of 512 × 512 [24]. Specifically, bilinear interpolation was used for original images, and the nearest neighbor interpolation method was used for liver and tumor labels. Every five slices were saved into a 3D patch and values from 0.5 to 99.5% were retained to obtain better contrast within the liver. and then normalized from 0 to 1. Various online data augmentation methods such as flip, rotate, and crop were performed during training [25]. Liver segmentation was performed using CSR-UNet to obtain the liver region; the main purpose of this step was to remove extra-hepatic interference information. The predicted liver region was regarded as the initial ROI for tumor coarse segmentation by 3D CSR-UNet model. Specifically, the segmented liver data were partitioned into multiple patches of size 192 × 192 × 80 and normalized before training [26]. Tumor coarse segmentation was trained using adaptive approach, and the prediction was performed by means of a sliding window with the patch overlapping mechanism. The results of multiple patches were assembled into an overall coarse segmentation result. Through this tumor coarse-segmentation module, most of the tumors larger than 30 mm could be segmented.

In order to improve the segmentation accuracy of small objects, detection module and CSR-UNet are added to the architecture to refine small tumor segmentation. The detection module based on nnDetection model [27] is used to detect only the space-occupying tumors smaller than 30 mm. nnDetection is a novel detection approach that employs Retina U-Net network [28], exhibiting exceptional segmentation performance in complex image scenes, with high precision and accuracy. In this step, tumors smaller than 30 mm were chosen for model training and validation. The detected tumors were cropped into a uniform size of 64 × 64 × 5 before inputting into the CSR-UNet network for tumor fine segmentation.

The final tumor segmentation result is a combination of the coarse-to-fine segmentation results. In specific, large tumor segmentation is achieved by coarse segmentation, while for small tumors, the initial ROI is further reduced with the help of a detection module, and fine segmentation is achieved. By fusing the results of coarse segmentation and fine segmentation, the final segmentation results of liver tumors can be obtained.

#### 2.2.2. CSR-UNet

Figure 4 shows the network architecture of CSR-UNet. CSR-UNet incorporates CSR module into the commonly used UNet framework. The difference between 3D and 2.5D CSR-UNet networks is whether the input is a 3D or a 5-layer 2D matrix, and the former uses 3D convolution while the latter is 2D convolution. The CSR module comprises dual 3 × 3 convolutional block with batch normalization, combined with an SE module and a residual module [29], as shown in Figure 3 on the left top. Batch normalization is the regularization during training to reduce generalization errors and overfitting. The main function of the SE module is to increase the attention perception of important regions, while the residual module reduces the difficulty of deep network training and improves the segmentation accuracy.

#### 2.2.3. Preprocessing and Adaptive Network Parameters

In order to obtain better network performance, a sliding augmentation module, random online augmentation module, and adaptive network parameters module are added to the CSR-UNet framework. These three modules are used to preprocess the data and generate the hyperparameters of the network before inputting them into the network. In the sliding augmentation module, we performed a sliding window crop for each input data matrix in each dimension [29]. The weights of the cropped input images are also different, giving higher weights to the positions corresponding to the images and labels that the network needs to focus on. The random online augmentation module is image augmentation of the input data, and different image augmentation methods are randomly occurring. These image augmentation methods include image rotation, image flipping, and image scaling. The adaptive network parameters module can consider the performance of the whole network, including the configuration of different patch sizes, learning rates, batch-size settings, etc. Through the complementary roles of these three modules, the data is then fed into the configured network for training iterations.

#### 2.2.4. Loss Function

Focal Tversky Loss is used in our network. Tversky coefficient is a generalization of Dice and Jaccard coefficients [30]. Similar to Focal loss, which focuses on difficult examples by reducing the weight of easy-to-use or common losses, Focal Tversky Loss also tries to learn difficult examples such as in the case of small ROIs (regions of interest) with the help of parameter coefficients [31].

Focal Tversky Loss is an improved version of the Tversky loss function, which is designed to address the issue of class imbalance in medical image analysis. In many medical imaging tasks, such as liver cancer detection, the number of abnormal pixels may be much smaller than the number of normal pixels, making it challenging to train deep neural networks to accurately recognize these abnormalities. Focal Tversky Loss introduces a focus parameter that allows the network to concentrate on difficult samples during training, thereby improving its ability to recognize abnormal pixels in small liver cancers; moreover, the loss function can be adjusted to balance the impact of classification errors on the overall loss, which further enhances the model’s performance in liver cancer recognition. Overall, Focal Tversky Loss is suitable for addressing small liver cancer recognition problems, as it helps the network learn relevant features and improve accuracy and robustness.

#### 2.2.5. Implementation Details

Both CSR-UNet and nnDetection are with optimal hyperparameters computed by an adaptive framework. CSR-UNet is implemented in PyTorch and trained on a cluster with 8 NVIDIA A100 GPUs. The network is involved in two parts, one of which is liver segmentation. In this phase, we resampled the image aspect size to 512 × 512 and processed their spacing counterparts into iso-voxels. Then we performed image up–down restriction according to the distribution of MRI intensity and normalized all input images. During the training period, we used 512 × 512 × 5 multi-channel input patches (equivalent to the form of 2.5D) and 512 × 512 × 1 outputs. We performed online augmentation of the images of the input network by applying random axis mirror flips with probability 0.5 in all 3 axes and random clockwise and counterclockwise rotations of 20°. In addition to this, all images were scaled with random intensity in the range (0.9, 1.1). The batch size of each GPU was set to 16, the learning rate was set to 0.001, and a cosine annealing learning rate scheduler and five-fold cross-validation were used. On this server, after the adaptive network parameters module and experimental verification, the loss reduction effect under these hyperparameters was the best. The epoch was 300 and the total training time was 3 days. The other part is the fine segmentation of small tumors, which differs from the former only in that the input images were boxes detected by multilayer nnDetection and portioned into 64 × 64 sizes, and the other preprocessing and augmentation methods were the same. The total time spent on this part of training was 16 h.

## 3. Results

To evaluate the effectiveness of the multi-network fusion segmentation method for small objects segmentation, we compared the proposed method with nnU-Net and several other networks. All comparison methods are listed in Table 1, including single 3D U-Net, single nnU-Net, nnDetection plus nnU-Net, and nnU-Net segmentation plus nnDetection detection with fine segmentation by CSR-UNet (ours). The results were compared from multiple perspectives for each tumor occupancy, including overall segmentation and detection metrics, and metrics grouped by different size distribution. In addition, the number distribution by long-diameter size, tumor types, and tumor benignity or malignancy are also analyzed.

The metrics we evaluated for liver tumor segmentation include segmentation metrics: Dice, IOU, volumetric similarity (VS), F1-score, and detection metric recall. Table 1 presents the mean inference metrics of fusion models trained by three randomly selected test sets. The test set comprised 44 MRI data. The table summarizes the performance results of five experiments, namely CSR-UNet fusion network (ours), 3D U-Net segmentation alone, nnU-Net segmentation alone, nnU-Net segmentation plus nnDetection detection, and nnU-Net segmentation plus nnDetection detection plus CSR-UNet fine segmentation. All five experiments used the same MRI data preprocessing methodology as the proposed method and utilized their respective segmentation networks to isolate the liver and apply the same preprocessing on the liver region. And, all experiments test the same 44 test data. Our proposed multi-network fusion approach outperformed all other methods with the best overall segmentation metrics across all test sets.

We also conducted a tumor count analysis in the three randomly sampled test sets, categorized by long diameter and benign or malignant classification, as presented in Table 2. The table illustrates that the majority of liver tumors in the test sets were smaller than 30 mm, which is the primary focus of our study. In addition, we provide detailed information regarding tumor categories in the three test sets. While benign and malignant categories were relatively balanced, hepatic cysts constituted the largest tumor category. Our experimental investigation and quantitative metric calculations yielded segmentation and detection results for tumors of varying size distributions, which are summarized in Table 3. The presented table reveals that the detection and segmentation performance of liver tumors with long diameters of 5–10 mm yielded the lowest indexes, with Dice coefficients ranging between 0.418 and 0.532; however, for tumors with long diameters of 10–30 mm, all related indicators improved significantly, with a recall rate of one achieved for tumors larger than 30 mm across all five methods. All results metrics for our proposed method here were also the highest for liver tumors in the less than 30 mm range. Notably, the end-to-end segmentation technique, nnU-Net, demonstrated only marginal improvements over 3D UNet. In contrast, the multi-network fusion method, specifically CSR-UNet, yielded substantial improvements when compared to nnU-Net. Lastly, our proposed method demonstrated optimal performance across all metrics related to long-diameter detection and segmentation.

Upon scrutinizing the inference result graph and the ground truth, our proposed method exhibits a relatively comprehensive efficacy in detecting small hepatocellular carcinoma. Nonetheless, certain small or poorly defined tumors are characterized by suboptimal boundary conformity, thereby resulting in diminished Dice and VS (volumetric similarity) scores. Moreover, the inadequacy of contrast and visualization in some tumors during the delayed phase impeded their accurate detection, leading to a proportion of false negatives.

To provide a rigorous elucidation, we posit that the suboptimal conformity of boundaries in certain smaller or indistinct tumors reduces the concurrence of the Dice and VS metrics. Additionally, the suboptimal contrast and visualization of some tumors during the delayed phase engendered erroneous classifications, resulting in an appreciable number of false negatives.

## 4. Discussion

In this paper, we proposed a method for small liver tumor detection and segmentation. Raw data from hospital screenings were collected and annotated to analyze the DCE images through expert annotation of additional information during the delay period. To maximize tumor ground segmentation results, we integrated detection and coarse and fine segmentation modules into our multi-network fusion method. The detection network employed nnDetection, which is a versatile adaptive detection network capable of detecting small targets comprehensively. Meanwhile, for the segmentation network, we developed a CSR-UNet network architecture that achieves more precise segmentation results for small liver cancers through preprocessing and training tumors at different scales. The detection and segmentation of small liver tumors through medical imaging is a crucial aspect of early disease diagnosis and treatment in the clinical setting. By comparing the results obtained from the test set, it can be inferred that the proposed model has the potential to assist in the early diagnosis of small liver tumors by providing valuable reference values; thus, this model holds promise for clinical applications in the detection and segmentation of small liver tumors.

Upon scrutinizing the inference result graph and the ground truth, our proposed method exhibits a relatively comprehensive efficacy in detecting small hepatocellular carcinoma. Nonetheless, certain small or poorly defined tumors are characterized by suboptimal boundary conformity, thereby resulting in diminished Dice and VS scores. Moreover, the inadequacy of contrast and visualization in some tumors during the delayed phase impeded their accurate detection, leading to a proportion of false negatives. To provide a rigorous elucidation, we posit that the suboptimal conformity of boundaries in certain smaller or indistinct tumors reduces the concurrence of the Dice and VS metrics. Additionally, the suboptimal contrast and visualization of some tumors during the delayed phase engendered erroneous classifications, resulting in an appreciable number of false negatives. Figure 5 presents the outcomes of our proposed method for detecting small hepatocellular carcinoma (HCC) with a diameter of less than 30 mm. Notably, the edges of these small HCCs appear indistinct in the images, rendering it challenging to determine whether they are tumors; however, our method can identify the majority of these small tumors. The comparison between the predicted results and the ground truth indicates that the accuracy of the segmentation is suboptimal. The small size of the HCCs leads to a limited number of pixels, resulting in relatively low values of segmentation-related metrics.

Despite the promising results, there are limitations to our current method that require further improvements. Firstly, the proposed model was only tested on a specific MRI dataset. To validate its generality, multicenter external test sets containing enhanced CT images and other small objects need to be collected. Furthermore, since the distribution of different liver tumor types in the dataset is not well-balanced, collecting more varied tumor types is necessary for deeper evaluation. Secondly, although we identified that the delayed phase provides more pronounced visual effects for the five liver tumor types we studied, there is a possibility that more prominent tumor types in other phases were missed. Future research may involve analyzing MRI multimodal data to further examine these tumors. Thirdly, due to the multi-network fusion method’s use of inference results from four neural networks, obtaining final segmentation results may take longer. Therefore, this method may be more suitable for patients with liver disease who do not require immediate surgery or require long-term follow-up. Fourthly, we did not model the classification for tumor type by single-class segmentation, and the method’s sensitivity or specificity can only be statistically derived for certain species. Finally, there is some room for optimization in the three network frameworks we used. For example, CSR-UNet can enhance the fineness of some patch segmentation, and UHRNet [32] can optimize the decoder. Future research will explore various method combinations to improve small object segmentation accuracy and efficiency.

## 5. Conclusions

This research presents a highly effective method for segmenting liver tumors by fusing multiple networks. Even for small tumors, this approach achieves outstanding segmentation and detection performance, thanks to the complementary information provided by the different networks. The fusion network increases the performance of network training through adaptive network parameters. The loss function combines Focal loss and Tversky loss, enabling network learning to focus on and learn more detailed tumor feature information. Moreover, the segmentation results enable precise quantification of each tumor’s long diameter and volume, providing valuable information for hepatologists and surgeons to monitor the progression and response to the treatment of patients with liver tumors. Ultimately, this novel approach maximizes the potential of liver tumor segmentation to inform clinical decision-making and improve patient outcomes.

## Figures and Tables

**Figure 1 diagnostics-13-02504-f001:**
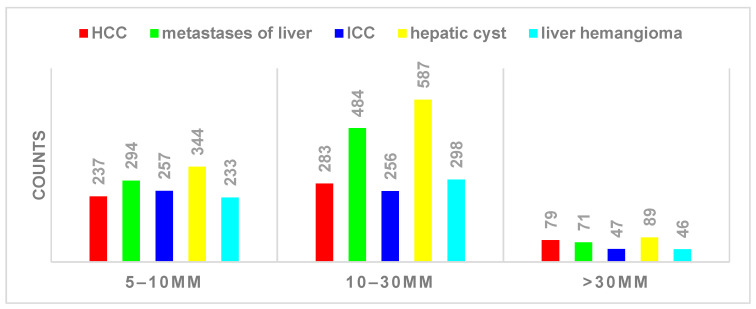
Class and size distribution of different tumors in the dataset.

**Figure 2 diagnostics-13-02504-f002:**
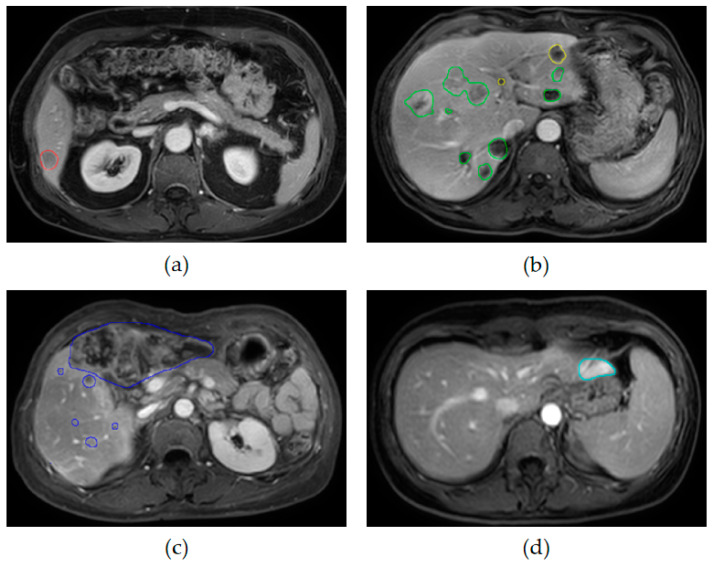
The five types of liver tumors with the ground truth: (**a**) HCC (red); (**b**) metastases of liver (green) and hepatic cyst (yellow); (**c**) ICC (dark blue); (**d**) liver hemangioma (light blue).

**Figure 3 diagnostics-13-02504-f003:**
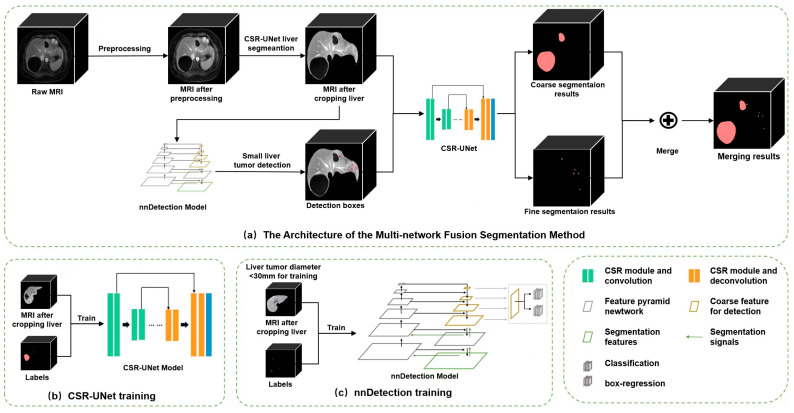
The architecture of the proposed coarse-to-fine fusion network. This present study utilized the raw MR images to extract the liver region via the implementation of a 3D CSR-UNet liver segmentation module. Following this, the region of interest (ROI) was subjected to two subsequent processes: Tumor coarse segmentation and tumor detection. The former was achieved through the use of a 3D CSR-UNet coarse segmentation module, whereas the latter was accomplished via training and inference with the Retina U-Net model utilizing the nnDetection detection module. The detected tumors were fine-segmented via the application of a 2.5D CSR tumor fine segmentation module and subsequently integrated with the coarse segmentation output, leading to the final segmentation results.

**Figure 4 diagnostics-13-02504-f004:**
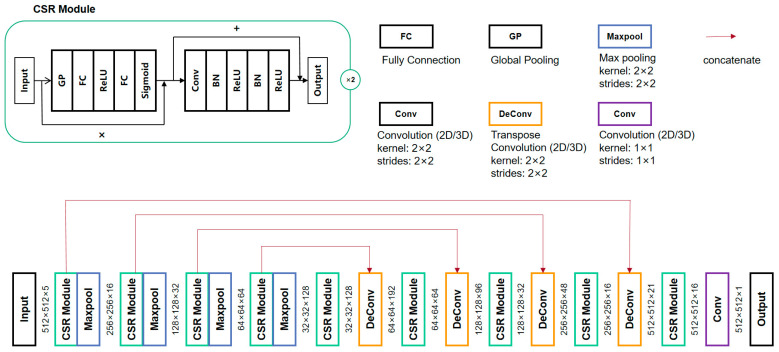
Overview of CSR-UNet in the coarse-to-fine fusion network.

**Figure 5 diagnostics-13-02504-f005:**
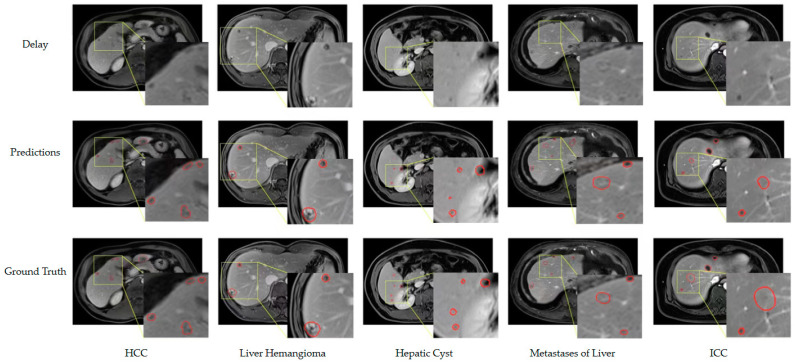
Typical results of our method on the test set. The MRIs are all delayed phases of DCE series. The predicted results and ground truth are circled with red edges in the figure.

**Table 1 diagnostics-13-02504-t001:** Overall segmentation performance of liver tumors by different methods.

Method	Dice	IOU	F1-Score	VS	Recall
3D U-Net	0.801	0.740	0.816	0.851	0.733
nnU-Net	0.847	0.750	0.823	0.859	0.776
nnDetection + nnU-Net	0.856	0.762	0.891	0.904	0.858
nnDetection + nnU-Net + CSR-UNet	0.859	0.771	0.895	0.904	0.858
**Proposed**	**0.869**	**0.778**	**0.897**	**0.910**	**0.867**

”VS” means “volumetric similarity”; “IOU” means “intersection over union”.

**Table 2 diagnostics-13-02504-t002:** Number (average) of size and type of liver tumors in the test set.

	Benign Tumor			Malignant Tumor	
Diameter (mm)	HCC	Metastases of Liver	ICC	Hepatic Cyst	Liver Hemangioma
5–10	24	83	60	105	31
10–30	84	85	74	130	78
>30	11	13	8	22	9

**Table 3 diagnostics-13-02504-t003:** Segmentation and detection metrics for tumor inference of different long diameters in our method and the comparison method.

Method	Diameter (mm)	Dice	IOU	F1-Score	Recall
3D U-Net	5–10	0.418	0.302	0.594	0.537
	10–30	0.520	0.382	0.829	0.847
	>30	0.821	0.749	1.000	1.000
nnU-Net	5–10	0.438	0.327	0.614	0.597
	10–30	0.521	0.430	0.869	0.857
	>30	0.842	0.791	1.000	1.000
nnDetection + nnU-Net	5–10	0.487	0.363	0.758	0.736
	10–30	0.531	0.398	0.910	0.905
	>30	0.857	0.807	1.000	1.000
nnDetection + nnU-Net + CSR-UNet	5–10	0.521	0.387	0.768	0.760
	10–30	0.552	0.448	0.910	0.760
	>30	0.873	0.829	1.000	1.000
**Proposed**	5–10	**0.532**	**0.409**	**0.785**	**0.761**
	10–30	**0.561**	**0.464**	**0.912**	**0.909**
	>30	**0.889**	**0.831**	1.000	1.000

## Data Availability

Data from this study were not provided in any public places. The code repository link https://github.com/wushu526/small_liver_tumor_segmentation (accessed on 14 June 2023).

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
