# Peer review of "A Coarse-to-Fine Fusion Network for Small Liver Tumor Detection and Segmentation: A Real-World Study"

_diagnostics, 2023, doi:10.3390/diagnostics13152504_

Round 1

Reviewer 1 Report

This paper is dealing with a coarse-to-fine fusion network for small liver tumor detection 2 and segmentation: a real-world study. The paper is well-written and organized. Maybe the conclusion should be rewritten to provide more information i.e. to be longer. 

Author Response

Thank you for your endorsement of this article. We have expanded the Conclusions section.

Reviewer 2 Report

Authors made a good attempt to segment the liver tumor from the MRI images.

This paper proposed a novel coarse-to-fine fusion segmentation approach to detect and segment small liver tumors with various sizes. To enhance the segmentation accuracy of small liver tumors, the method incorporates a detection module and a CSR (convolution-SE-residual) module, which includes a convolution block, a SE (squeeze and excitation) module, and a residual module for fine segmentation. However, few suggestions to enhance the submitted research article are given below.

1. The recent literatures which may support for the existing modules such as nnU-Net, nnDetection+nnU-Net and nnDetection + nnU-Net + CSR-UNet are not found anywhere in the paper. Authors are advised to include the relevant references.

2. Since, the the selected dataset has less number of image samples, it is suggested to augment the input image after pre-processing.

3. A separate subsection 1.1 can be included as research contributions.

4. The resolution of Fig 2,  3 and 4 can be improved.

5.  It is advised to included the training and testing accuracy and loss graph.

6. The authors need to justify the number of epochs, loss function chosen and step size.

7. Limitations of the proposed model is missing.

Minor edits are required.

Author Response

Thank you for recognizing and advising us on this article. In response to your suggestions, we have the following improvements:

  1. By querying, there is no method of combining nnU-Net with nnDetection in recent literature, and there is no method of combining nnU-Net, nnDetection and CSR-UNet. According to your suggestion, we have expanded some nnU-Net related information in 1.1.Related Work with, and the relevant content of nnDetection is mentioned in 2.2.1.Architecture.
  2. Data augmentation methods are mentioned in 2.2.2. Preprocessing and Adaptive Network Parameters, and we have expanded some specific methods about random online augmentation module.
  3. We added 1.1. Related Work to 1. Introduction.
  4. We uploaded higher resolution Fig 2, 3 and 4.
  5. Sorry we only have the training and validation loss graph of CSR-UNet and we lost other loss graph. The graph is not included in nnDetectionframework from the source code.
  6. We added some descriptions of epochs and step size in 2.2.4. Implementation Details. These parameters are empirical conclusions obtained from many experiments.The specific information of the loss function has been mentioned in 2.2.3. Loss Function.
  7. Limitations of the proposed model is in the third paragraph of 4. Discussion.

Reviewer 3 Report

The paper entitled "A coarse-to-fine fusion network for small liver tumor detection and segmentation: a real-world study" proposes a novel coarse-to-fine fusion segmentation method to detect and segment small liver tumors with various sizes. The authors' method incorporated in the segmentation process a supplementary module as convolution-SE-residual ; in this context, five types of liver tumors were tested.

1. Please add the section Related Work

2. For each liver tumor, add an image with its ground truth.

3. The 44 test sets feed five CNNs CSR-UNet fusion network, 3D U-Net segmentation, nnU-Net segmentation, nnU-Net segmentation plus nnDetection detection, and nnU-Net segmentation plus, it is not clear how many images belong each set.

4. In experiments, a private liver MRI dataset was used. Please add a link or upload the images and code of CNNS in a repository.

5. In the section "2.2.4. Implementation Details," it is unclear if the enumerated hyperparameters were proposed for each CNN or were changed.

6. The results are not compared with others from the scientific literature

7. The quality of the figures should be improved.

 8. For the hyperparameters, were they established default values? It is unclear if the authors used the ablation or tuning process; if not, add this information.

Author Response

Thank you for recognizing and advising us on this article. In response to your suggestions, we have the following improvements:

  1. We added 1.1. Related Work to 1. Introduction.
  2. We added the 5 kinds of liver tumors with their ground truth in Figure 2.
  3. It is a mistake and that is not ‘44 test sets’. We have a test set which has 44 MR images.
  4. Since the source of the data set is the hospital, it is not convenient to disclose it, so we can only share the code repository https://github.com/wushu526/small_liver_tumor_segmentation
  5. In 2.2.4. Implementation Details, we added a description of some hyperparameters obtained through the adaptive network parameters module and experimental verification.
  6. We have compared the classical method nnU-Net with 3DU-Net in experiments. Especially nnU-Net, which is already a benchmark comparison and segmentation method, so we did not add other comparison methods from scientific literature.The 3D U-Net method is already well known in the industry. Therefore, the specific method is not mentioned in the article, but this method is used in the comparison.
  7. We uploaded higher resolution Fig 2, 3 and 4.
  8. These hyperparameters are obtained through the adaptive network parameters module and experimental verification. In Table 3., the displayed content can represent the results of ablation experiments. The separate nnU-Net results, the results of nnU-Net after passing the nnDetection detection, and the results of the small target segmentation of nnDetection detection plus nnU-Net plus CSR-UNet, are all consistent with this method (nnDetection+CSR-UNet ) for comparison.

Round 2

Reviewer 2 Report

Authors made a good attempt to address the queries raised in the previous version. It can be accepted

Reviewer 3 Report

The paper can be published in its present form.